# Anti-Fouling Performance of Hydrophobic Hydrogels with Unique Surface Hydrophobicity and Nanoarchitectonics

**DOI:** 10.3390/gels8070407

**Published:** 2022-06-27

**Authors:** Liangpeng Zeng, Ziqi Liu, Jingliang Huang, Xiaolin Wang, Hui Guo, Wei-Hua Li

**Affiliations:** 1School of Chemical Engineering and Technology, Sun Yat-sen University, Zhuhai 519082, China; zenglp6@mail2.sysu.edu.cn (L.Z.); liuzq3355@gmail.com (Z.L.); huangjliang3@mail.sysu.edu.cn (J.H.); 2School of Materials Science and Engineering, Sun Yat-sen University, Guangzhou 510275, China; 3State Key Laboratory of Quality Research in Chinese Medicine and School of Pharmacy, Macau University of Science and Technology, Taipa 999078, Macao

**Keywords:** water content, hydrophobic hydrogels, surface hydrophobicity, anti-fouling

## Abstract

Hydrogel is a kind of soft and wet matter, which demonstrates favorable fouling resistance owing to the hydration anti-adhesive surfaces. Different from conventional hydrogels constructed by hydrophilic or amphiphilic polymers, the recently invented “hydrophobic hydrogels” composed of hydrophobic polymers exhibit many unique properties, e.g., surface hydrophobicity and high water content, suggesting promising applications in anti-fouling. In this paper, a series of hydrophobic hydrogels were prepared with different chemical structures and water content for anti-fouling investigations. The hydrophobic hydrogels showed high static water contact angles (WCAs > 90°), indicating remarkable surface hydrophobicity, which is abnormal for conventional hydrogels. Compared with the conventional hydrogels, all the hydrophobic hydrogels exhibited less than 4% *E. coli* biofilm coverage, showing a contrary trend of anti-fouling ability to the water content inside the polymer. Typically, the poly(2-(2-ethoxyethoxy)ethyl acrylate) (PCBA) and poly(tetrahydrofurfuryl acrylate) (PTHFA) hydrogels with relatively high surface hydrophobicity showed as low as 5.1% and 2.4% *E. coli* biofilm coverage even after incubation for 7 days in bacteria suspension, which are about 0.32 and 0.15 times of that on the hydrophilic poly(*N*,*N*-dimethylacrylamide) (PDMA) hydrogels, respectively. Moreover, the hydrophobic hydrogels exhibited a similar anti-adhesion ability and trend to algae *S. platensis*. Based on the results, the surface hydrophobicity mainly contributes to the excellent anti-fouling ability of hydrophobic hydrogels. In the meantime, the too-high water content may be somehow detrimental to anti-fouling performance.

## 1. Introduction

Biofouling, caused by the attachment and colonization of micro-organisms onto the surfaces submerged in the aqueous environments, leads to the deterioration of marine equipment and structures and modification of surface roughness, which has a serious negative impact both on people’s daily marine activities and on the marine industry [1,2]. Consequently, the development of materials with fouling-resistant properties that could eliminate the issue resulting from marine biofouling is feasible and urgently needed. Recently, many anti-fouling strategies of utilizing coating materials to prevent protein, cell, and fouling organism adhesion have been proposed, including (super)hydrophobic materials [3,4], bio-inspired antiadhesive micro/nanostructures [5,6], siloxane/fluorocarbon-based fouling release coatings [1,7], amphiphilic coatings [8,9], and polymer gels [10,11,12,13,14]. In particular, hydrogels, a soft and wet polymeric material, are significantly different from the dry and hard solid surfaces, which play crucial roles in the anti-fouling property due to the potential super wettability and surface lubricity [15].

Hydrogels are defined as a kind of novel semi-solid material composed of crosslinked three-dimensional polymer networks and water [16]. Benefitting from the unique composition and structure, hydrogels often display lots of fascinating performances, such as super wetness, softness, and biocompatibility [17,18,19]. All these characteristics endue the hydrogels with multitudinous functionality and applications, which have attracted great attention [20,21,22,23]. More importantly, the trapped plenty of water and its natural fluidity also confer a hydration layer to the surface of the hydrogel, resulting in the liquid-like property of hydrogels. Based on this particularity, hydrogels are also commonly used as anti-fouling materials to resist biological adhesion. For example, Jiang et al. developed a series of hydrogels with zwitterionic polymer for anti-fouling applications [24,25]. The robust liquid-like surfaces endowed by the hydration layer resulted in excellent underwater superoleophobicity and anti-adhesion performance toward various fouling organisms. In addition to the stable hydration layer, Gong et al. proposed the contributions of low elastic modulus resulted from the super hygroscopicity of the polymer network on the anti-fouling properties of hydrogels [26,27].

Indeed, the low adhesion between the unwanted biofouling and the substrate′s surface is primarily dependent on the low elastic modulus and low surface free energy [28]. For the conventional hydrogels, the trapped water layer will be easily damaged in dynamic and harsh environments and fail to reduce the inhibition of the biofouling. How to maintain the low elastic modulus and low surface energy of hydrogels is the key to its anti-fouling ability. Fortunately, the recently invented “hydrophobic hydrogels” provide an ideal choice for anti-fouling hydrogels [29,30]. The “hydrophobic hydrogels” are composed of hydrophobic polymer networks while maintaining super-high water content [29]. Moreover, more importantly, the hydrophobic hydrogels possess a unique fruit-like structure, in which the semipermeable skin layer formed in the swelling process provide the surface hydrophobicity and anti-drying property [17,29]. This combination of the softness and high water storage of conventional hydrogels, together with the low surface free energy of siloxane/fluorocarbon-based fouling release coatings, may enable the hydrophobic hydrogels to act as the most promising anti-fouling hydrogels.

In this work, we fabricated a series of hydrophobic hydrogels, composed of a hydrophobic polymer with different structures and water content, by immersing hydrophobic organo-gels with omniphilic dimethyl sulfoxide (DMSO) in water. Based on the unique dynamic solvent-exchange processes, the water content inside the hydrogels was adjustable and controlled by the swelling process in the DMSO [17]. The systematical investigations on the anti-fouling performance of various hydrophobic hydrogels were conducted in this work, including polymer chemical structure, water-content factor, and surface characteristics. In contrast to the typical hydrophilic hydrogels, the hydrophobic hydrogels displayed distinct hydrophobic surface properties and exhibited excellent biofouling resistance against the adhesion of bacteria (*Escherichia coli*) and algae (*Spirulina platensis*). It is concluded that the superior anti-fouling ability of hydrophobic hydrogels mainly depended on their surface hydrophobicity. Moreover, the surface characteristic shows a certain polymer structure effect. This work offers a unique strategy to develop hydrogel coatings with persistent anti-biofouling properties and provides a new potential application for hydrophobic hydrogels.

## 2. Results and Discussion

### 2.1. Swelling Performance

According to our previous study, different from conventional hydrogels, the swelling behavior of hydrophobic hydrogels in water goes through a unique process of first swelling and then slowly shrinking with the increasing of swelling time. It exhibited a dynamic swelling behavior and obvious time dependence on the immersion process in water. In addition, their swelling behavior in water is also affected by the nature of the solvent inside the gels’ network. The omniphilic organic solvents with a high affinity for water could result in the super-high swelling of the gels, while the solvent exchange occurs once the gels are immersed in water due to the driving force of the osmotic pressure. Therefore, DMSO was selected as the organic solvent for hydrophobic hydrogel in the preparation and swelling process for exchange with water. Benefitting from the unusual solvent-exchange process, it is feasible to prepare the hydrophobic hydrogel with defined water content by controlling the content of initial organic solvent DMSO inside gels.

The organo-gels with the different chemical structures were prepared by solvent polymerization with the defined DMSO content of 79.70 wt% initiated by UV light. As depicted in Figure 1, the organo-gels exhibited complete transparency after preparation and subsequent swelling in DMSO for 36 h. During the immersion in a large amount of DMSO, most of the as-prepared discoid hydrophobic organo-gels were significantly swollen in the first 3 h and gradually reached an equilibrium state after tens of hours. In the meantime, the volumes of these organo-gels boosted up to two times with swelling in DMSO from the previous as-prepared state. Notably, the conventional hydrophilic poly(*N*,*N*-dimethylacrylamide) (PDMA) and poly(acrylamid) (PAAm) organo-gels expand more rapidly than the hydrophobic ones and reached an equilibrium state within 6 h in DMSO. Interestingly, the hydrophobic poly(2-(2-ethoxyethoxy)ethyl acrylate) (PCBA) organo-gels also exhibit similar swelling behaviors in DMSO. Followed with immersion in water, the hydrophobic pre-equilibrated organo-gels with DMSO changed from transparent to opaque white within a few minutes. It indicated that the solvent exchange successfully occurred in water and led to the instantaneous phase separation of hydrophobic polymer chains. After a few days’ solvent-exchange process, the DMSO inside the hydrophobic organo-gels was completely exchanged into the water due to the high osmotic pressure formed by the absorbed DMSO. All the hydrophobic gels significantly expanded again and reach a maximum state with the exchange of water. Macroscopically, the resulting hydrophobic hydrogels’ volume was boosted 1–2 times from equilibrated organo-gels with DMSO (Figure 1). In comparison with the hydrophobic gels, the analogous phenomenon was not observed in the hydrophilic control (PAAm and PDMA gels), which maintains the almost original transparency and size. According to the previous study [29], a probable semipermeable membrane, resulted from hydrophobic-polymer-chain phase separation, formed and resulted in the unique water-swelling behavior.

Although all hydrophobic gels exhibit similar swelling behavior in water, there are some obvious differences among gels with different chemical structures. As shown in Figure 2, the PCBA organo-gels composed of polymers with hydrophilic poly(ethylene glycol) (PEG) side chains tended to swell rapidly and reached an equilibrium state in DMSO. After immersing in a large amount of water, the solvent exchange completed as quickly as that of hydrophilic PDMA and PAAm organo-gels. As a comparison, the poly(benzyl acrylate) (PBnA) and poly(2-phenoxyethyl acrylate) (PPHEA) organo-gels, of which the polymer has bulk aryl hydrophobic groups, demonstrated a longer swelling process prior to the equilibrium state in DMSO. The poly(methyl acrylate) (PMA) organo-gels took the longest swelling time to the maximum swelling degree. After immersing in water, the solvent-exchange process of other organo-gels, except for the PCBA, PDMA, and PAAm, becomes more time-consuming. The behavior can be explained by the slower release of DMSO inside caused by the formed more solvent-proof semi-permeable membranes due to the stronger hydrophobicity than the others [17]. During this process, the poly(phenyl acrylate) (PPA) organo-gels pre-equilibrated with DMSO changed from soft to brittle and plastic. Consequently, the surface topography of PPA hydrogels became rather uneven. This may be caused by the PA linear polymer with a rather high glass-transition temperature (T_g_ = 63 °C) compared with those of other conventional and hydrophobic gels. It can be seen that both the swelling behavior in DMSO and solvent exchange in water, as well as the water content of hydrophobic hydrogels, displayed a significant polymer effect.

### 2.2. Surface Hydrophobicity

From the screen of the actual photos, we can roughly see that the surface of the resulting hydrophobic hydrogel is relatively smoother than the control. To demonstrate the estimate, the wettability experiments of all hydrogels were performed by static contact angle measurements. As shown in Figure 3, all the hydrophobic hydrogels exhibited surface hydrophobicity; this, however, was not exactly the same as that of various hydrogels with different chemical structures and water content. It can be seen that the static water contact angles (WCAs) of the PDMA and PAAm hydrogels are all less than 55°, indicating strong surface hydrophilicity. Nevertheless, unlike the conventional hydrogels where the water inside the polymer network interacted with the polymer chain ends, a hydrophobic interaction formed on the surface of the gels composed of hydrophobic polymer chains. Consequently, the WCAs of all hydrophobic hydrogels obtained by direct water exchange after polymerization revealed more than 90° when 3 μL water drops fall on their surfaces, which is much higher than the mentioned hydrophilic hydrogels (Figure 3a). In contrast to the common hydrogels, the intrinsic hydrophobic nature of polymer chains makes a decisive impact on the high water contact angles (hydrophobicity). The polymer content also plays a role in the hydrophobicity of hydrophobic hydrogels (Figure 2). Moreover, the measured water contact angles of hydrogels may be affected by their surface topographies, resulting from the differences in the hydrophobic group of polymer chains. The WCA of the exceptional PCBA hydrogels exhibited a super high 104°, which contributed to the rather rapidly formed smooth semi-permeable membrane surface due to the shortest solvent-exchange process.

Similarly, as displayed in Figure 3b, all hydrophobic hydrogels except for the PPHEA and poly(tetrahydrofurfuryl acrylate) (PTHFA) obtained by the water exchange of organo-gels with equilibrium state in DMSO demonstrated high hydrophobicity but yielded slightly higher WCAs than those of ones obtained by direct water exchange. When the trapped water inside the hydrophobic polymer chains increased, the surface hydrophobic performance of hydrophobic hydrogels was more evident, due to the lesser amount of the relative polymer content and the resulted close-knit hydrophobic dense surface skin.

In addition, the surfaces of some gels (i.e., PPA and PBnA hydrogels) became macroscopic concave–convex after the solvent-exchange process in water. To confirm the surface structure, the 3D stereoscopic microscope was applied to characterize the surface morphologies of hydrogels obtained by the solvent exchange of organo-gels with an equilibrium state in DMSO. As you can see in Figure 4, in contrast to the hydrophilic ones, the hydrophobic gels exhibited distinct surface roughness after the water–DMSO exchange process in water. Typically, it can be seen that the PPA hydrogels show the most significant surface undulation, followed closely by PBnA and PPHEA hydrogels, suggesting the noticeable surface roughness of hydrophobic hydrogels. In reality, the unevenness on the outside surface of PPA hydrogels was also observed from the macroscopic view (see Figure 2). These phenomena and trend predicted the surface hydrophobicity of hydrophobic hydrogels. However, the surface roughness did not completely conform to the trends of WCAs. In addition to the intrinsic nature of polymer chains and polymer content, the unique surface topographies of hydrophobic hydrogels inevitably affect the WCAs on their surfaces, which give hydrophobic hydrogels surface hydrophobicity.

### 2.3. Anti-fouling Performance

To evaluate the anti-fouling property of the as-obtained hydrophobic hydrogels, anti-bacterial and anti-algae adhesion experiments were carried out. Figure 5 and Figure 6 present the results of the adhesion tests of *Escherichia coli* (*E. coli*) on the hydrophobic and conventional hydrogels after 2 and 7 days, respectively. As shown in Figure 5a, the bacteria *E. coli.,* adhering to conventional PDMA and PAAm hydrogels, exhibited as low as 4.8% and 5.8% biofilm coverage, suggesting a more than 50% decrease compared with the surface of the glass control (that of 13.5%) after incubation for 48 h in *E. coli* suspension. By contrast, all hydrophobic hydrogels obtained by direct water exchange after polymerization exhibited fewer bacterial adhesion amounts than conventional hydrogels. In particular, the PCBA and PTHFA hydrogels showed 1.02% and 0.72% *E. coli* biofilm coverage, which are about 0.22 and 0.15 times of that on the PDMA hydrogels, respectively. The *E. coli* attachments on these hydrogels were consistent with the trend of their surface hydrophobicity; that is, the PCBA and PTHFA hydrogels with higher WCAs revealed the lowest *E. coli* attachment. As shown in Figure 5b, the results of the anti-fouling test on the hydrogels obtained by the water exchange of organo-gels with equilibrium state in DMSO also displayed a superior anti-adhesion ability to *E. coli*. The hydrophobic hydrogels exhibited less than 2% *E. coli* biofilm coverage and suggested a general decreased tendency compared with those obtained by direct water exchange after polymerization. It indicated the anti-fouling of *E. coli* increased with the increase of surface hydrophobicity but decreased with the increase of water content inside the polymer. It is concluded that the excellent anti-fouling property of hydrophobic hydrogels is attributed to its strong surface hydrophobicity arising from specific hydrophobic polymer network structures and hydrophobic dense surface skin. Moreover, after incubation for 7 days, the bacterial *E. coli* coverages on the surfaces of typical PCBA, PTHFA, and PPHEA hydrogels obtained by the water exchange of organo-gels with equilibrium state in DMSO increased to 5.1% and 2.4%, which are approximately 0.32 and 0.15 times of that on the PDMA hydrogels, respectively (Figure 6). It demonstrated a slight increase with these typical hydrophobic hydrogels, whereas the glass and hydrophilic hydrogels control exhibited sharply increased bacterial adhesion. It suggests that the hydrophobic hydrogels with extremely high water content have long-term anti-bacterial adhesion stability.

In addition to the anti-bacterial adhesion, the hydrophobic hydrogels exhibited excellent anti-fouling ability to the algae *Spirulina platensis* (*S. platensis*). Figure 7 shows the biofilm coverage and microscopy images of algae biofouling on the control and the three typical hydrophobic hydrogels with different water content after 7 days of immersion in the *S. platensis* culture medium at 30 °C, respectively. Compared with hydrophilic hydrogels and the glass control, the three typical hydrophobic hydrogels with low water content showed less than 10% biofilm coverage, suggesting the excellent algae anti-fouling performance (Figure 7a). As shown in Figure 7b, the anti-algae adhesion performance of the three hydrophobic hydrogels with high water content was higher than that of ones with low water content. However, the *S. platensis* biofilm coverage was still far below those on the control surfaces, showing an excellent fouling resistance to algae, anti-fouling as expected. In summary, the anti-algae adhesion performance was nearly consistent with the results of anti-bacterial adhesion.

Generally, the anti-fouling ability of conventional hydrogels mainly depends on their surface hydration layer. It means that the water-swelling ability is crucial to the anti-fouling ability. In this case, hydrophobic hydrogels possess a stronger water-swelling capacity than hydrophilic hydrogels by the means of solvent exchange. It is obvious that the water content of hydrophobic hydrogels also greatly depends on the DMSO content in the gel network (Figure 2). More interestingly, all hydrophobic hydrogels obtained by either direct water–DMSO exchange, or solvent exchange after swelling equilibrium with DMSO, exhibited ultra-high water content and unique surface hydrophobicity, which is completely different from hydrophilic hydrogels (Figure 3). This characteristic may result from the hydrophobic nature of the polymer and the resulting surface topography and structure (i.e., surface roughness and semipermeable dense skin). As a result, hydrophobic hydrogels have excellent anti-fouling performance.

According to the theory proposed by et al. [28], it can be explained in two ways. First and foremost, the special chemical character of the polymer and resulting surface hydrophobicity of constructed hydrogels play important roles in inhibiting fouling attachment. The surface hydrophobicity quantified by high WCAs is generally considered a feature of low surface free energy, which makes a direct connection to the low adhesion strength to fouling organisms (or a contaminant) [7,8]. Clearly, the hydrophobic hydrogels present extremely favorable conditions for low surface free energy. Despite the surface of hydrogels being uneven, the hydrophobic hydrogels exhibited more significant anti-fouling ability than hydrophilic hydrogels. On the other hand, the hydrophobic hydrogels possess extremely high water absorption and retention to enable and maintain softness and wetness, resulting in the low elastic modulus for a low adhesion mechanism similar to common hydrogels.

## 3. Conclusions

In this work, we fabricated a series of hydrophobic hydrogels composed of hydrophobic polymers with different chemical structures through a solvent-exchange method using organo-gels infiltrated with DMSO. The water content inside the hydrophobic hydrogels was adjustable and controlled by this solvent-exchange manner. The hydrophobic hydrogels with different water contents showed extremely high static contact angles, indicating remarkable surface hydrophobicity different from conventional hydrogels. The hydrophobic hydrogels showed excellent anti-fouling ability to resist Gram-negative bacteria *E. coli* and algae *S. platensis*. Compared with the common hydrophilic hydrogels, surface hydrophobicity is the main contribution to the excellent anti-fouling performance. More importantly, with the aid of low elastic modulus resulting from high water absorption and retention inside polymer networks, the hydrophobic hydrogels reveal greater potential than common hydrophilic hydrogels in anti-fouling applications in the future.

## 4. Materials and Methods

### 4.1. Materials

*N*,*N*-dimethylacrylamide (DMA), acrylamide (AAm), methyl acrylate (MA), ethyl acrylate (EA), phenyl acrylate (PA), 2-(2-ethoxyethoxy)ethyl acrylate (CBA), 2-phenoxyethyl acrylate (PHEA), and photo-initiator 2-oxoglutaric acid (α-keto) were obtained from the Shanghai Aladdin Biochemical Technology Co., Ltd., Shanghai, China. Benzyl acrylate (BnA) and tetrahydrofurfuryl acrylate (THFA) were purchased from the Shanghai Macklin Biochemical Co., Ltd., Shanghai, China. Crosslinking agents 1,4-butandiol diacrylate (BDA) were provided by 3A Chemicals. Dimethyl sulfoxide (DMSO) was analytical grade and water was purified with a Millipore system, combining inverse osmosis membrane and ion exchange resins for synthesis, purification, and the swelling test.

### 4.2. Preparation of Hydrogels

First, the organo-gels were prepared by photochemical initiated polymerization in the solvent, DMSO. The initial monomer concentration was kept as 20 wt%, and the amounts of crosslinking agent, BDA, and photo-initiator, α-keto, were fixed at 1 mol% and 0.2 mol% to monomer, respectively. Typical PMA organo-gels were taken for an example to illustrate the preparation of the organo-gels. MA monomer, 1 mol% of BDA, and 0.20 mol% of α-keto with respect to MA, were dissolved in DMSO solution (MA concentration equal to 20 wt% of the solution) and deoxygenated under an argon atmosphere. Under an argon atmosphere, the solution was poured at room temperature between two glass plates, separated by a silicone rubber spacer with thicknesses of 1 mm. The resulting solution system was carried out by ultraviolet photopolymerization (wavelength ≈ 365 nm) for 6 h at 25 °C. Then, the as-prepared organo-gels, after polymerization, were immersed in a large amount of DMSO, or water, to obtain gels with defined states.

All hydrophobic and hydrophilic hydrogels are obtained by the solvent-exchange procedure from DMSO in the organo-gels networks to the water. On the one hand, the as-prepared organo-gels, after solvent polymerization with DMSO, were directly immersed in a large amount of pure water. The water was exchanged once per 8 h to obtain hydrogels with a thermodynamic equilibrium state. On the other hand, the as-prepared organo-gels after solvent polymerization with DMSO were immersed in a large amount of DMSO for 1.5 days. During this process, DMSO was exchanged once per 12 h to reach a thermodynamic equilibrium state and wash away any residual chemicals. The resulting organo-gels, equilibrated with DMSO, were immersed in pure water for at least one week with a frequent water exchange.

### 4.3. Characterizations

Swelling tests: The as-prepared organo-gels were cut into a defined shape with discoid geometry of the initial diameter (d) = 32 mm. During the solvent (DMSO or water) swelling process, the weights of all samples were recorded at a specific time. The weight of swollen gels in aqueous media (*m*_a_) and the gels with organic solvent before water immersion (*m*_o_) were obtained. Water content of the gels was determined as the percentage weight change of the organo-gels: *q* = [(*m*_a_ − *m*_o_)/*m*_a_] × 100%, where *m*_o_ is the mass of the organo-gels with DMSO before water immersion, and *m*_a_ is the mass of water-swollen gels after a specific time in the water. The surface water of the swollen gels was carefully wiped using dust-free tissue paper and *m*_a_ was measured. At least three distinct assays were performed to obtain the mean value and standard deviation.

All the optical pictures presented in this study were taken by a digital camera (Canon-EOS 80D).

Surface wettability characterization of the hydrogels: The static water contact angles (WCAs) were evaluated by an optical contact angle measuring instrument (OCA15EC, Dataphysics, Filderstadt, Germany). Deionized water (3 μL) was used as the probing liquid for the measurements. For each sample, the mean value was obtained by measuring at least three different positions.

Surface morphologies characterization: The 3D stereoscopic microscope (Leica DVM6, Wetzlar, Germany) was applied to characterize the surface morphologies of hydrogels.

Mechanical characterization: Tensile tests of hydrogels were conducted on a commercial tensile test machine equipped with a 50 N load cell (Model AGX-V 50N, Shimadzu Co., Ltd., Kyoto, Japan). The samples were pre-cut into dumb-bell shape with a width of 1.6 mm and an initial gauge length of 15 mm. The thickness of the specimens was measured before every test. All tests were performed at room temperature (~25 °C) with a stretch rate of 100 mm/min.

### 4.4. Anti-fouling Test

Bacterial biofouling test: The fluorescently tagged *Escherichia coli* (*E. coli*) were used as the model bacteria for the bacterial attachment experiments for this study. After an 18 h incubation, the viable cell numbers of the bacterial suspension were on the order of 10^8^ mL^−1^ (OD = 1). After sterilization by UV radiation for 30 min, the glass slide and experimental samples were exposed to the 150-times diluted *E. coli* in Luria–Bertani (LB) broth, and cultured on a rocking table with a shaking speed of 120 rpm at 37 °C for 48/168 h. Following exposure to the bacterial suspensions, each specimen was gently rinsed in phosphate-buffered saline (PBS) three times and dried for 30 min at room temperature. Afterward, the samples were observed with the bacteria attached to the surfaces by fluorescence microscopy (excitation wavelength of 460 nm–550 nm and emission wavelength of 590 nm, N-STORM, Nikon, Tokyo, Japan). The *E.*
*coli* biofouling was reported as biofilm coverage of cells that was calculated by the software *ImageJ* (Java 1.8.0_172 [64-bit]).

Algae biofouling test: *Spirulina platensis* (*S. platensis*) was obtained from the commercial Guangzhou Youbei Biotechnology Co., Ltd., Guangdong, China, and cultured in Zarrouk medium with glucose, appended under a light incubator (light intensity of 4000 lx) at 30 °C for 7 days. 20 mL *S. platensis* log-phase suspension was diluted 10 times with the medium and allowed to incubate for the algae biofouling test. After 7 days at 30 °C, each sample was gently washed in deionized water to remove cells that had not been attached to the surface. The cell attachment and biofilm growth were characterized using a fluorescence microscope (N-STORM, Nikon, Tokyo, Japan). The algae biofouling was reported as the biofilm coverage of cells that was calculated by the software *ImageJ* (Java 1.8.0_172 [64-bit]).

## Figures and Tables

**Figure 1 gels-08-00407-f001:**
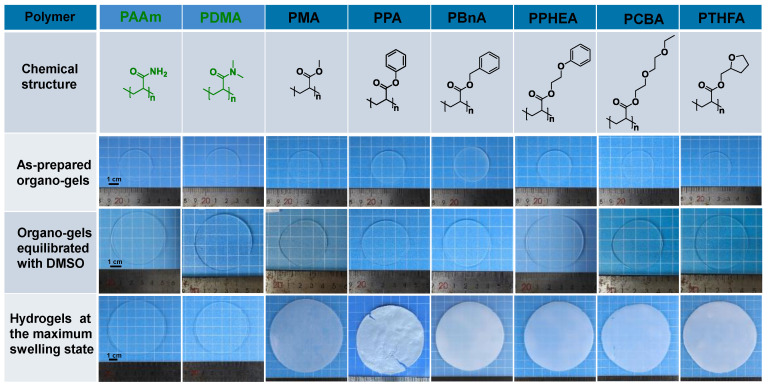
The chemical structure of the polymers for conventional and hydrophobic hydrogels, and photos of the resulting organo-gels at different stages and hydrogels after the solvent exchange in water to reach equilibrium states.

**Figure 2 gels-08-00407-f002:**
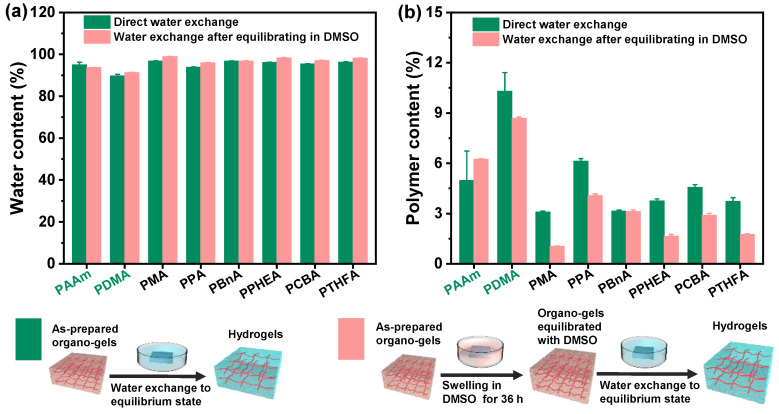
The water content and polymer content of the hydrogels with the different chemical structures obtained by two different routines of the direct solvent exchange of the as-prepared organo-gels after polymerization and the solvent exchange of organo-gels with equilibrium state in DMSO, respectively: (**a**) the water content of hydrogels; (**b**) the polymer content of hydrogels.

**Figure 3 gels-08-00407-f003:**
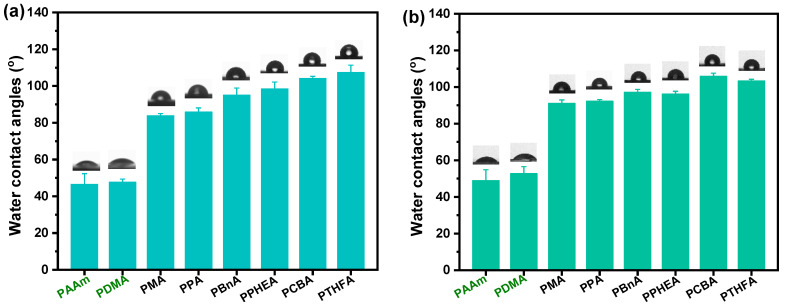
The static water contact angles of the hydrogels with the different chemical structures and water content: (**a**) various hydrogels obtained by the direct solvent exchange of as-prepared organo-gels after polymerization; (**b**) various hydrogels obtained by the solvent exchange of organo-gels with equilibrium state in DMSO.

**Figure 4 gels-08-00407-f004:**
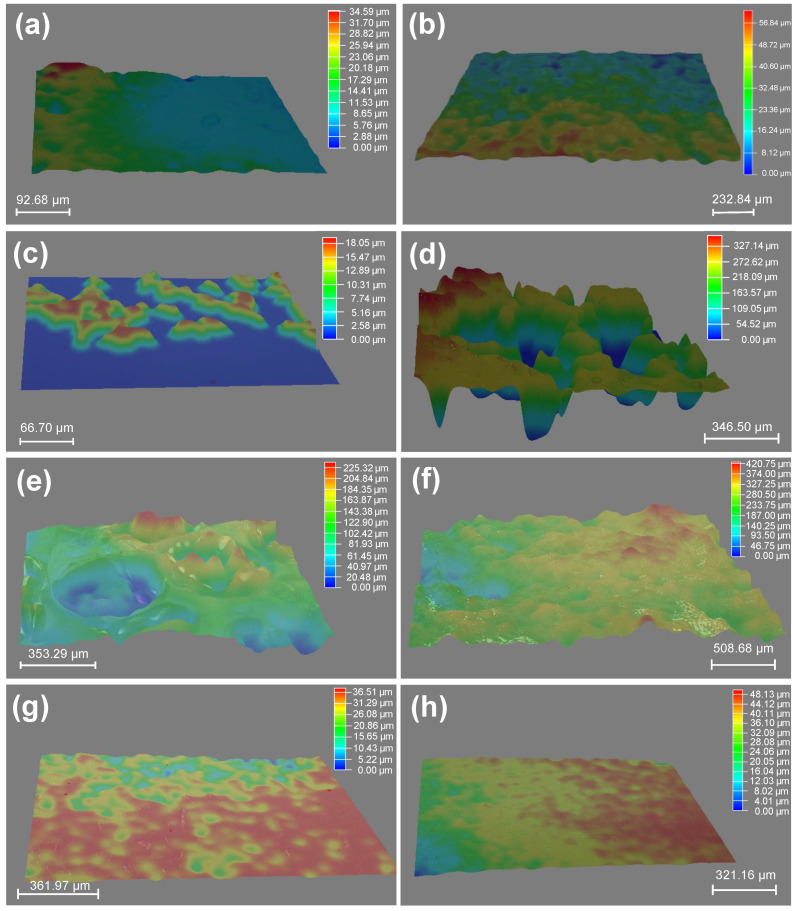
3D images of surface morphologies of hydrogels obtained by the solvent exchange of organo-gels with equilibrium state in DMSO: (**a**) PAAm; (**b**) PDMA; (**c**) PMA; (**d**) PPA; (**e**) PBnA; (**f**) PPHEA; (**g**) PCBA; (**h**) PTHFA.

**Figure 5 gels-08-00407-f005:**
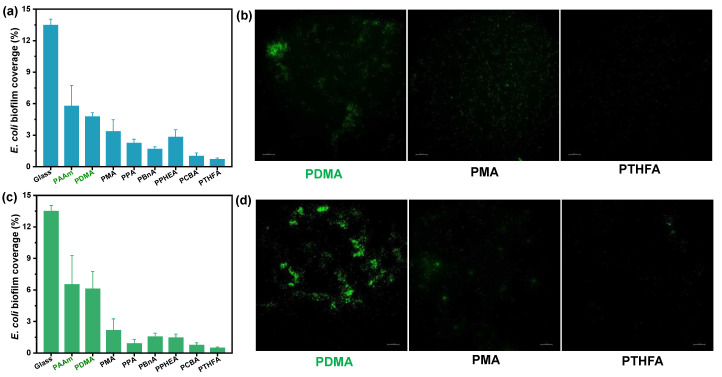
*E.* coli bacteria biofouling evaluation of typical hydrogels obtained by (**a**,**b**) the direct solvent exchange of as-prepared organo-gels after polymerization, and (**c**,**d**) the solvent exchange of organo-gels with equilibrium state in DMSO after incubation for 48 h in *E. coli* suspension. (**a**,**c**): Image analysis results (% biofilm coverage) of various surfaces: statistical analysis of the mean fluorescence intensity. (**b**,**d**): Typical fluorescence microscopy images of algae biofouling on hydrogels.

**Figure 6 gels-08-00407-f006:**
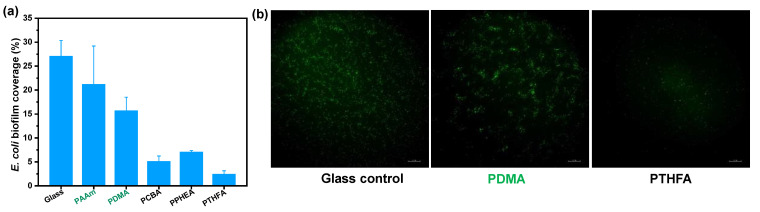
*E. coli* bacteria biofouling evaluation of typical hydrogels obtained by the solvent exchange of organo-gels with equilibrium state in DMSO after incubation of 168 h in *E. coli* suspension. (**a**) Image analysis results (% biofilm coverage) of various surfaces: statistical analysis of the mean fluorescence intensity. (**b**) Typical fluorescence microscopy images of *E. coli* biofouling on hydrogels.

**Figure 7 gels-08-00407-f007:**
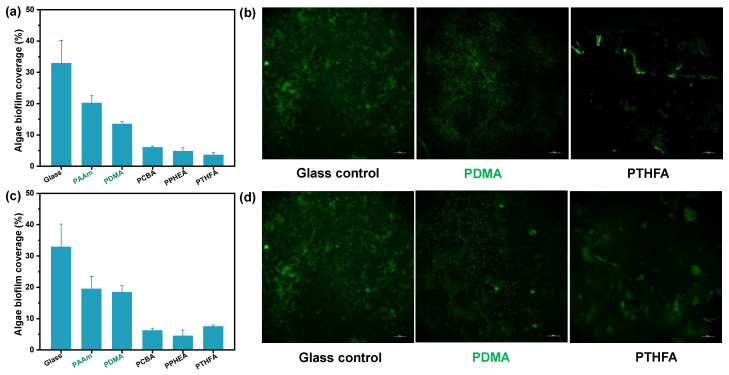
Algae biofouling evaluation of typical hydrogels obtained by (**a**,**b**) the direct solvent exchange of as-prepared organo-gels after polymerization, and (**c**,**d**) the solvent exchange of organo-gels with equilibrium state in DMSO after incubation of 168 h in *S. platensis* suspension. (**a**,**c**): Image analysis results (% biofilm coverage) of various surfaces: statistical analysis of the mean fluorescence intensity. (**b**,**d**): Typical fluorescence microscopy images of algae biofouling on glass substrate and hydrogels. Note that the glass control in Figures b and d are same.

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
