# Peer review of "Anti-Fouling Performance of Hydrophobic Hydrogels with Unique Surface Hydrophobicity and Nanoarchitectonics"

_gels, 2022, doi:10.3390/gels8070407_

Round 1

Reviewer 1 Report

This work investigated the anti-fouling performance of a series of hydrophobic hydrogels with varied chemical structures. Compared with conditional hydrophilic hydrogels, the hydrophobic hydrogels exhibit remarkable excellent performance. The manuscript is suggested to be published. However, the reviewer found the following sections are hard to follow:

  1. The authors mentioned that the anti-fouling hydrogels usually have low elastic modulus. But in some applications, such as marine equipment, the anti-fouling materials should meet the requirement of high toughness and strength. What is the elastic modulus and toughness of these hydrophobic hydrogels? And how to resolve the conflict between low elastic modulus (high anti-fouling) and high toughness?
  2. In Figs. 2,3,4,5,6, the results of direct water exchange and water exchange after equilibrating state in DMSO are shown. But in Fig. 1, the author only shows the photos of samples obtained by water exchange after equilibrating state in DMSO. It would be better also to show the photos of samples obtained by direct water exchange. So, the readers can catch the differences between samples fabricated by different routines intuitively.
  3. The authors should check more carefully on “as-prepared” and “as-obtained”, several misuses are found, such as on Page 4, line 17 and line 24.
  4. Although the chemical structures of the samples are shown in Fig.1, it would be better to give the whole name of the chemicals when mentioned in the main text for the first time.
  5. In hydrogel preparation, the authors sometimes described the chemical content by weight fraction, sometimes by molar fraction. It would be better to use terminology consistently.
  6. In the swelling test, the m0 first denotes “the weight of the gels with organic solvent before water immersion”, and later denotes “the mass of the original polymer after polymerization”. The original polymer usually stands for dry polymer not the gels, please confirm it.
  7. The exchange time of water-DMSO is only 1.5 days. In such a hydrophobic and dense skin, whether the DMSO has been completely dialyzed out? How did the authors analyze whether it still contains DMSO or not after solvent exchange? Because the DMSO may affect the bacteria's survival, the authors should pay attention to it.
  8. “It is worth noting that the contact angles of these hydrophobic hydrogels present an increasing trend with the hydrophobic group of polymer chains”. What is the sequence order of the hydrophobic groups? Whether the groups are more hydrophobic, the contact angels are larger? If not, what is the reason?
  9. In real application cases, the anti-bacterial and anti-algae performances should maintain for at least several months. What are the long-term anti-bacterial and anti-algae performances of these hydrophobic hydrogels?

Reviewer 2 Report

Although some data were obtained, necessary data and discussions are not well included. I may recommend publication of this work in Gels but some revisions are necessary. Please see below.

1) Clearer representation on quantitative relation between major parameters, Antifouling performance, swelling performance, and surface hydrophobicity, have to be given and discussed in details.

2) The current title is too simple and cannot give insightful impression. Addition of a new conceptual term often changes such information. I may suggest use on an emerging conceptual term, nanoarchitectonics, in the title (as post-nanotechnology concept, see https://pubs.rsc.org/en/content/articlelanding/2021/NH/D0NH00680G). For example, the title like Nanoarchitectonics for Anti-fouling Performance of Hydrophobic Hydrogels ... may sound more attractive.

3) Characterization (confirmation) data of chemical structures have to be given (IR spectra ???).

4) Reference selection is not bad, but more general comprehensive papers can be cited. Recent comprehensive and general papers on hydrogels had better be cited more (for example, https://www.journal.csj.jp/doi/10.1246/bcsj.20210209, https://www.sciencedirect.com/science/article/pii/S0079642520300669?via%3Dihub, https://pubs.acs.org/doi/10.1021/acsnano.1c04206, https://www.journal.csj.jp/doi/10.1246/bcsj.20210234)

5) Morphological data of gel structures (SEM ???) have to be given.

Reviewer 3 Report

This article is interesting and innovative. However even if the biofilm formation assays are well performed, the characterization of the gels should be significantly improved.

*Some SEM images of the surfaces of the hydrogels before and after swelling could be performed ...or alternatively some characterization with AFM. Indeed the surface roughness of the swollen hydrogels could modify the value of the static water contact angles.

*Some allusions are made on the low Young moduli of those hydrogels. What are the orders of magnitude ? Is there a variation with the kind of used monomers ?

*Concerning the quantification of biofilms using fluorescence, some more experimental details should be provided: excitation and emission wavelength ?

Some spelling mistakes: 

*end of page 2: "driving force of the osmotic pressure"

*page 4: "relatively smoother than the control"

*page 4, legend of Figure 2 "the as obtained hydrogels" is repeated twice.

Round 2

Reviewer 2 Report

Replies and revisions are fine. The revised version becomes acceptable.

Reviewer 3 Report

The authors replied to my comments in a satisfactory manner. This manuscript can now be accepted for publication.